# Aberrant information transfer interferes with functional axon regeneration

**Chen Ding[1], Marc Hammarlund[1,2]\***

[1]Department of Neuroscience, Yale University, New Haven, United States;
[2]Department of Genetics, Yale University, New Haven, United States

**Abstract** Functional axon regeneration requires regenerating neurons to restore appropriate synaptic connectivity and circuit function. To model this process, we developed an assay in *Caenorhabditis elegans* that links axon and synapse regeneration of a single neuron to recovery of behavior. After axon injury and regeneration of the DA9 neuron, synapses reform at their pre-injury location. However, these regenerated synapses often lack key molecular components. Further, synaptic vesicles accumulate in the dendrite in response to axon injury. Dendritic vesicle release results in information misrouting that suppresses behavioral recovery. Dendritic synapse formation depends on dynein and *jnk-1*. But even when information transfer is corrected, axonal synapses fail to adequately transmit information. Our study reveals unexpected plasticity during functional regeneration. Regeneration of the axon is not sufficient for the reformation of correct neuronal circuits after injury. Rather, synapse reformation and function are also key variables, and manipulation of circuit reformation improves behavioral recovery.
DOI: https://doi.org/10.7554/eLife.38829.001

## Introduction

Axon injury disconnects neurons from their postsynaptic targets and destroys circuit functions and behavioral outputs. Axon regeneration—in which injured neurons initiate growth, find appropriate targets, and reconnect functional circuits—has the potential to restore function after nerve injury. Many studies have demonstrated that some behavioral recovery can occur after axon injury. For example, *Caenorhabditis elegans* are able partially to recover locomotion after transection of multiple GABA motor neurons (*Byrne et al., 2016*; *El Bejjani and Hammarlund, 2012*; *Yanik et al., 2004*). Fish recover swimming behavior rapidly following spinal cord injury (*Becker et al., 1997*; *Bernstein, 1964*; *Briona and Dorsky, 2014*; *Davis and McClellan, 1993*; *Oliphint et al., 2010*). Further, in many cases, improved recovery after axon injury has been correlated with increased axon regeneration. Increased regeneration in *C. elegans* after treatment with a PARP inhibitor correlates with improved behavioral recovery (*Byrne et al., 2016*). In the mouse spinal cord, various approaches to improve axon regeneration also result in increased behavioral recovery (*Bradbury et al., 2002*; *Kim et al., 2004*; *Ramer et al., 2000*). Similarly, in the optic nerve, regeneration is increased and partial visual function restored by co-deletion of PTEN and SOCS3 together with application of 4-AP (*Bei et al., 2016*; *Thanos et al., 1997*). These studies and others establish a link between axon regeneration and behavioral recovery.

Injured neurons that contribute to behavioral recovery must do so by rewiring into relevant circuits. Consistent with this, regenerated axons have been observed to form new synapses. For example, when injured rat retinal ganglion axons are given permissive conduits that allow regeneration, these neurons can re-establish presynaptic specializations after regenerating into the superior colliculus (*Vidal-Sanz et al., 1991*; *Vidal-Sanz et al., 1987*). Similarly, neurons in the optic nerve of goldfish and giant reticulospinal axons of larval lamprey are also able to regenerate synapses (*Meyer and Kageyama, 1999*; *Oliphint et al., 2010*; *Wood and Cohen, 1981*). In the rat peripheral

**\*For correspondence:**
marc.hammarlund@yale.edu

**Competing interests:** The authors declare that no competing interests exist.

nervous system, regenerated preganglionic axons have also been shown to form new synapses at the originally denervated postsynaptic sites (*Raisman, 1977*). However, although these morphological and behavioral data suggest that regenerated neurons do form synapses and rewire into circuits, it has not been possible to analyze whether and how individual regenerated neurons contribute to circuit function and behavioral recovery.

Here, we describe a new in vivo model for functional regeneration that allows analysis of how individual regenerated neurons rewire into circuits and drive behavior. Using the DA9 motor neuron of *C. elegans,* we show that even after axons regenerate and reform synapses, regenerated neurons recover only a fraction of their original circuit function. A key limitation on functional recovery is misrouted information transfer from regenerated neurons, due to ectopic synapse formation in the dendrite. Preventing ectopic synapse formation by eliminating the MAP kinase *jnk-1* improves functional recovery. Together, our experiments establish the landscape for functional recovery of neurons after regeneration, and show that manipulations that correct the function of regenerated circuits result in improved behavioral recovery after nerve injury.

## Results

### Formation of normal and dendritic synapses after regeneration of a single neuron

DA9 is a bipolar excitatory motor neuron, the most posterior neuron of the DA class, with its cell body in the ventral midline. According to the electron microscopy reconstruction, the DA9 dendrite extends anteriorly in the ventral nerve cord (VNC), receiving input from command neurons and sensory neurons (*Hall and Russell, 1991b*; *White et al., 1976*; *White et al., 1986*) (*Figure 1A*). The DA9 axon extends posteriorly in the VNC and then crosses to the dorsal nerve cord (DNC) via a commissure. It then proceeds anteriorly in the DNC and forms en passant cholinergic synapses that are dyadic with the posterior dorsal body wall muscles (BMWs) and the ventral D (VD) inhibitory GABA motor neurons. These synapses are restricted to a small region of the DA9 axon, and the axon itself extends in the DNC past the synaptic region. Based on its anatomy and connectivity, DA9

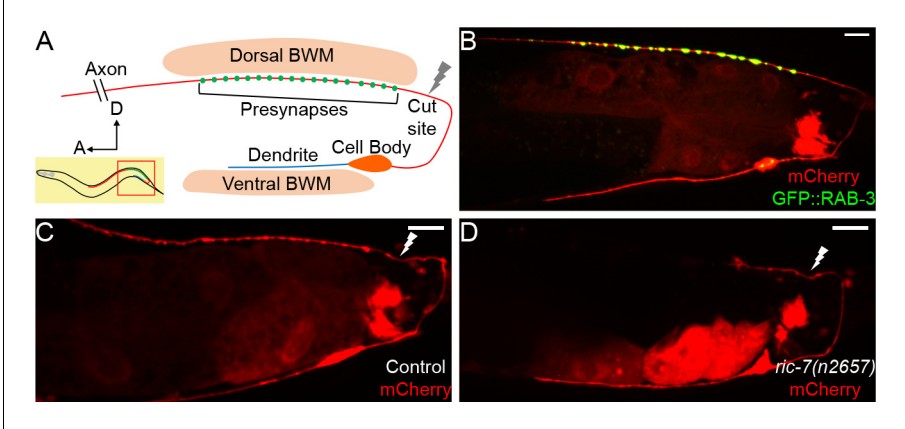

**Figure 1.** DA9 axon regeneration and degeneration. (**A**) Diagram of DA9 morphology and the laser cutting site. (**B**) A control adult expressing GFP::RAB-3 and mCherry in DA9. mCherry expression is also observed in the posterior gut in these micrographs. (**C**) A control animal 12 hr after axotomy. The distal axon fragment remains and overlaps with the regenerating axon. The white lightning mark indicates the axotomy site. (**D**) A *ric-7(n2657)* animal 12 hr after axotomy. The distal fragment has degenerated. Scale bars = 10 μm.
DOI: https://doi.org/10.7554/eLife.38829.002

The following figure supplement is available for figure 1:

**Figure supplement 1.** *ric-7(n2657)* animals show normal DA9 development, better regeneration and increased degeneration of GABA motor neurons.
DOI: https://doi.org/10.7554/eLife.38829.003

activity is predicted to trigger dorsal bending of the animal's tail region, by directly activating dorsal BMWs while simultaneously inhibiting opposing ventral BMWs via VD activity.

To analyze the morphology of DA9 neurites and synapses in the context of axon regeneration, we used the DA9-specific *itr-1 pB* promoter to co-express soluble mCherry along with the presynaptic marker GFP::RAB-3, as previously described. In the posterior of the animal, this promoter drives expression exclusively in the DA9 neuron, as well as in the posterior cells of the intestine (*Klassen and Shen, 2007*) (*Figure 1B*). These labels allow us to observe the DA9 dendrite, axon, and synaptic vesicle clusters, and are consistent with previous observations and with the electron microscopy reconstruction (*Hall and Russell, 1991b*; *Klassen and Shen, 2007*; *White et al., 1976*). We used a pulsed laser essentially as described (*Byrne et al., 2011*) to sever the DA9 axon just posterior to its synaptic region in the DNC. We severed axons in L4 stage animals and assessed regeneration. We found that 12 hr after axotomy, DA9 axons had initiated regeneration and regenerated past the injury site (*Figure 1C*), similar to the kinetics of regeneration in other neurons (*Chuang et al., 2014*; *Hammarlund and Jin, 2014*; *Yanik et al., 2004*). However, even after 12 hr, the distal axon fragment still was present (*Figure 1C*), similar to slow removal of fragments after injury in other *C. elegans* neurons (*Nichols et al., 2016*). Presence of this distal fragment raises the possibility that regenerating DA9 axons may restore connectivity simply by fusion with the fragment, as previously observed in *C. elegans* (*Abay et al., 2017*; *Ghosh-Roy et al., 2010*; *Neumann et al., 2015*; *Neumann et al., 2011*). Alternatively, the fragment may interfere with synaptic regeneration by occupying relevant anatomical sites. By contrast, distal axon fragments degenerate rapidly in vertebrates and *Drosophila* through a process called Wallerian degeneration (*MacDonald et al., 2006*; *Martin et al., 2010*; *Waller, 1850*). In these cases, successful degeneration is thought to be permissive for axons to regenerate and re-innervate the targets. Indeed, delaying Wallerian degeneration has been shown to impair regeneration and delay the locomotor recovery of mice (*Bisby and Chen, 1990*; *Brown et al., 1994*; *Zhang et al., 1998*).

In order to build a permissive environment for synaptic regeneration and block fusion, we used a recently identified mutation, *ric-7(n2657)*, to accelerate degeneration of DA9 distal axon fragments after injury (*Hao et al., 2012*; *Nichols et al., 2016*; *Rawson et al., 2014*). DA9 develops normally in *ric-7(n2657)* animals compared to controls (*Figure 1—figure supplement 1A–1D*), consistent with the previous data (*Hao et al., 2012*). However, after laser axotomy of DA9, the distal axon segment degenerated quickly and was largely cleared away at 12 hr (*Figure 1D*). In the GABA motor neuron system, *ric-7(n2657)* animals also show enhanced degeneration together with increased regeneration 24 hr after axotomy, indicating that loss of *ric-7(n2657)* and enhanced degeneration do not interfere with axon regeneration (*Figure 1—figure supplement 1E–1H*). These experiments indicate that *ric-7(n2657)* eliminates potential interference from the remaining distal axon fragments after axotomy, facilitating the study of new synapse formation during axon regeneration. Therefore, we included the *ric-7(n2657)* mutation in the following experiments unless further mentioned.

We characterized the characteristics and kinetics of DA9 axon regeneration and synapse reformation by examining animals at different time points after axotomy (*Figure 2A–2D*). Axon growth occurred only in the first 48 hr, after which no additional growth occurred (*Figure 2G*). During this time, regenerating axons completely reinnervated their previous synaptic region, but failed to grow far past the synaptic region as observed before injury (*Figure 2D and G*; the grey stripe in *Figure 2G* indicates the synaptic region of regenerated axons 48 hr after axotomy). Lack of growth into the distal asynaptic area resulted in regenerated axons that only reach 1/3 of length of intact axons (*Figure 2G*, cut 72 hr vs uncut 48 hr, mean = 182.0 μm and 518.2 μm, p<0.0001, unpaired t test). Overall, 100 percent (28/28) of severed DA9 axons initiated regeneration and reinnervated the synaptic region. The high incidence of regeneration and accurate growth to the former synaptic area we observed is ideal for analyzing the function of regenerated neurons, and is likely due to our choice of injury site close to the DNC. By contrast, if DA9 was severed at the dorsal-ventral midline, it often failed to reinnervate the synaptic area (*Figure 2—figure supplement 1*).

As DA9 regenerates, it forms new SV puncta coincident with regeneration (*Figure 2B–2E*). A prominent feature of DA9 synapses in intact neurons is their stereotypic placement along the DNC, in which there is a specific synaptic region containing a stereotyped number of puncta that occupies a limited and stereotyped region of the axon (*Figure 1A and B*) (*Hall and Russell, 1991b*; *Klassen and Shen, 2007*; *Poon et al., 2008*). Regenerated neurons are largely able to restore the stereotyped arrangement of SV puncta. By 48 hr, when axon growth was completed (*Figure 2G*),

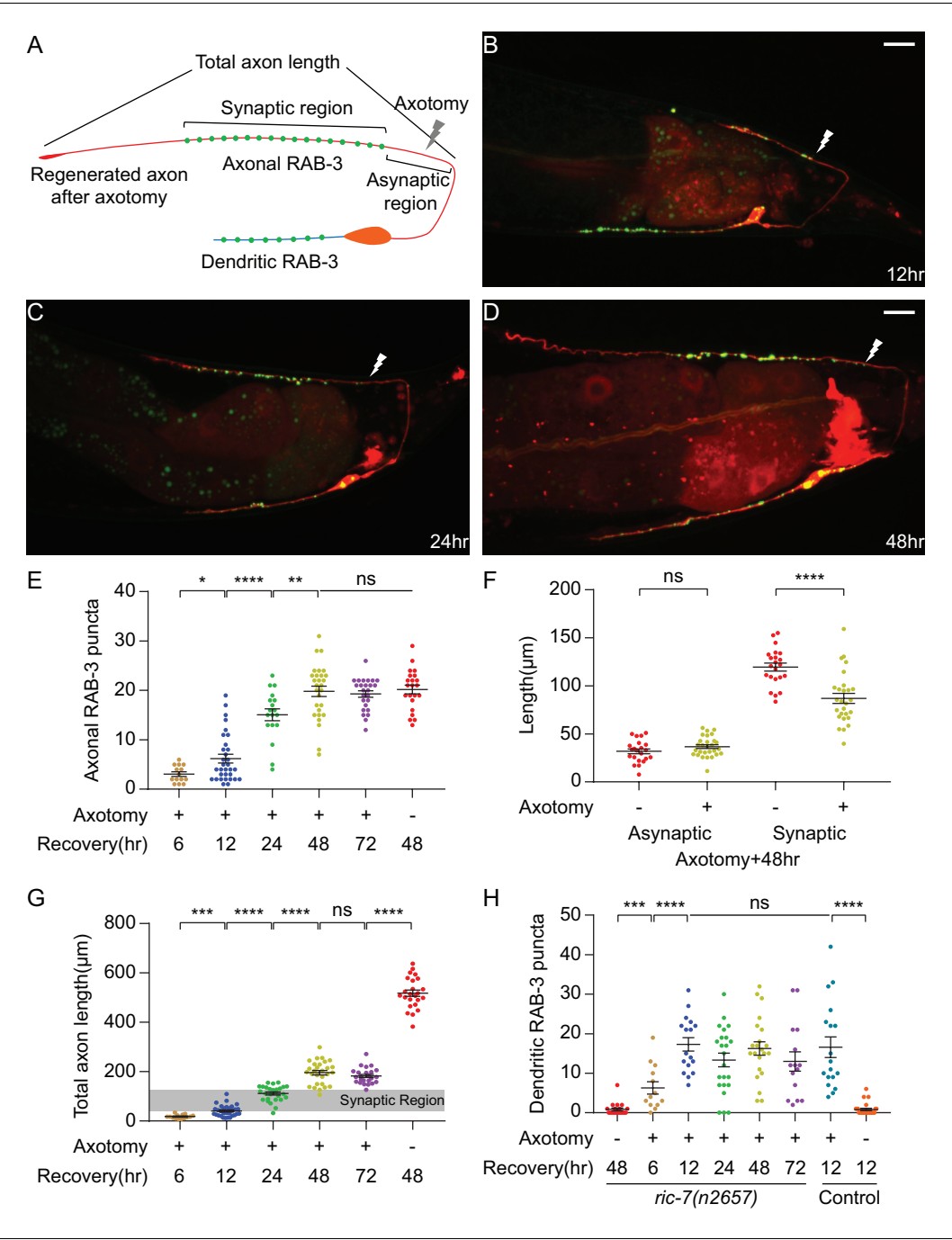

**Figure 2.** DA9 forms normal and dendritic synapses during regeneration. (**A**) Diagram of a regenerated DA9 axon. (**B–D**) DA9 axonal and synaptic regeneration at different timepoints after axotomy. The white lightning mark indicates the axotomy site. Scale bars = 10 μm. (**E**) Number of axonal RAB-3 puncta at different timepoints with or without axotomy. Mean ± SEM. *p<0.05; **p<0.01; ****p<0.0001; ns, not significant. Unpaired t test for comparisons between two groups and one-way ANOVA for more than two groups. (**F**) Length of the asynaptic and the synaptic region in intact and axotomized animals 48 hr after axotomy. Mean ± SEM. ****p<0.0001; ns, not significant. Unpaired t test. (**G and H**) Total axon length and number of dendritic RAB-3 puncta at different timepoints with or without axotomy. Mean ± SEM. ***p<0.001; ****p<0.0001; ns, not significant. Unpaired t test for comparisons between two groups and one-way ANOVA for more than two groups.

DOI: https://doi.org/10.7554/eLife.38829.004

The following figure supplements are available for figure 2:

**Figure supplement 1.** Axotomy in the DNC enables accurate axon growth following the original path.

*Figure 2 continued on next page*

*Figure 2 continued*

DOI: https://doi.org/10.7554/eLife.38829.005

**Figure supplement 2.** Regeneration of other presynaptic components.

DOI: https://doi.org/10.7554/eLife.38829.006

the number of SV puncta was statistically indistinguishable from intact animals and did not further increase (*Figure 2E*, cut 48 hr, cut 72 hr and uncut 48 hr, mean = 19.8, 19.3 and 20.2, p=0.7864, one-way ANOVA). The new synapses were again localized in a specific region of the DNC (*Figure 2D*). The position of the new synaptic region, as indicated by the length of the asynaptic region (*Figure 2A*), was similar to intact animals (*Figure 2F*, uncut vs cut, mean = 31.8 μm and 36.7 μm, p=0.1238, unpaired t test). The largest difference was in the length of the synaptic region, which was shorter than in uninjured axons (*Figure 2F*, uncut vs cut, mean = 119.5 μm and 87.0 μm, p<0.0001, unpaired t test). Overall, these data indicate that the regenerating DA9 axon is able to form new SV puncta and largely re-establish its synaptic pattern. The process of axon regeneration, both in terms of axon regrowth and synapse reformation, ended by 48 hr after axotomy.

Normally, multiple protein components localize to active zones at the presynaptic terminal, where SV clusters are found. To determine whether the new SV clusters in the regenerated axons are fully reconstructed pre-synapses, we analyzed two active zone proteins, UNC-10 and CLA-1S. UNC-10 is the *C. elegans* homolog of vertebrate Rim1, which regulates the priming step of vesicle fusion (*Koushika et al., 2001*). CLA-1S is the small isoform of a recently identified protein that is homologous to vertebrate active zone proteins Piccolo and Bassoon (*Xuan et al., 2017*). In intact animals, UNC-10 and CLA-1S puncta were predominantly localized to puncta in the DA9 synaptic region (*Figure 2—figure supplement 2A–2A''* and 2 C-2C''). However, unlike SVs, UNC-10 and CLA-1S puncta regenerated only partially in the new DA9 synaptic region 48 hr after axotomy (*Figure 2—figure supplement 2B–2B''* and 2D-2D''). The number of UNC-10 and CLA-1S puncta after axotomy was about half the number in intact animals (*Figure 2—figure supplement 2E and F*). These data suggest that some of the new SV clusters in the regenerated axons represent defective synapses, and indicate that mechanisms for regenerating SV clusters and active zones are at least partially distinct.

Strikingly, SV puncta also formed in the dendrite during regeneration (*Figure 2A–2D*). In intact animals, SV puncta were essentially absent from the dendrite (*Figure 1B*). After axon surgery, however, dendritic SVs became apparent at as early as 6 hr after axotomy (*Figure 2H*). The number of puncta reached a maximum at 12 hr (mean = 17.3) and decreased only slightly thereafter. Importantly, the appearance of ectopic SVs in the dendrite was not caused by the *ric-7(n2657)* background, because control animals also accumulated a similar number of ectopic puncta in response to axotomy (*Figure 2H*, *ric-7(n2657)* cut 12 hr vs control cut 12 hr, mean = 17.3 and 16.6). These data show that axonal injury triggers ectopic accumulation of SVs in the dendrite, identifying an unexpected form of plasticity during recovery from nerve injury.

## Postsynaptic receptors maintain their localization and are aligned with regenerated DA9 SV clusters after axotomy

Postsynaptic neurotransmitter receptors are normally juxtaposed to SV clusters and active zones in the presynaptic neuron, facilitating neurotransmission. To test if postsynaptic receptors are correctly localized after regeneration, we labeled ACR-16 in the body wall muscles and ACR-12 in the GABAergic motor neurons. ACR-16 is a nicotinic acetylcholine receptor (nAChR) subunit in muscles (*Francis et al., 2005*), and ACR-12 is a nAChR subunit expressed in the major classes of motor neurons (*Petrash et al., 2013*). ACR-16 in the posterior dorsal BWMs and ACR-12 in the posterior VD GABA neurons are expected to be postsynaptic to dyadic DA9 presynapses in the dorsal nerve cord, according to the anatomy and previous studies (*Hall and Russell, 1991b*; *Klassen and Shen, 2007*; *White et al., 1986*). In intact animals, both ACR-16::GFP and ACR-12::GFP puncta overlapped with the putative DA9 synaptic region (*Figure 3A*, *Figure 3—figure supplement 1A–1A''*). 48 hr after axotomy, the number of puncta in this region was unchanged (*Figure 3B–3C*, *Figure 3—figure supplement 1B–1C*). In addition, the length of the region of the DA9 axon in the posterior dorsal nerve cord that is not apposed to ACR-12 or ACR-16 puncta was also unchanged 48 hr after axotomy (*Figure 3D*, *Figure 3—figure supplement 1D*). These data indicate that the number and

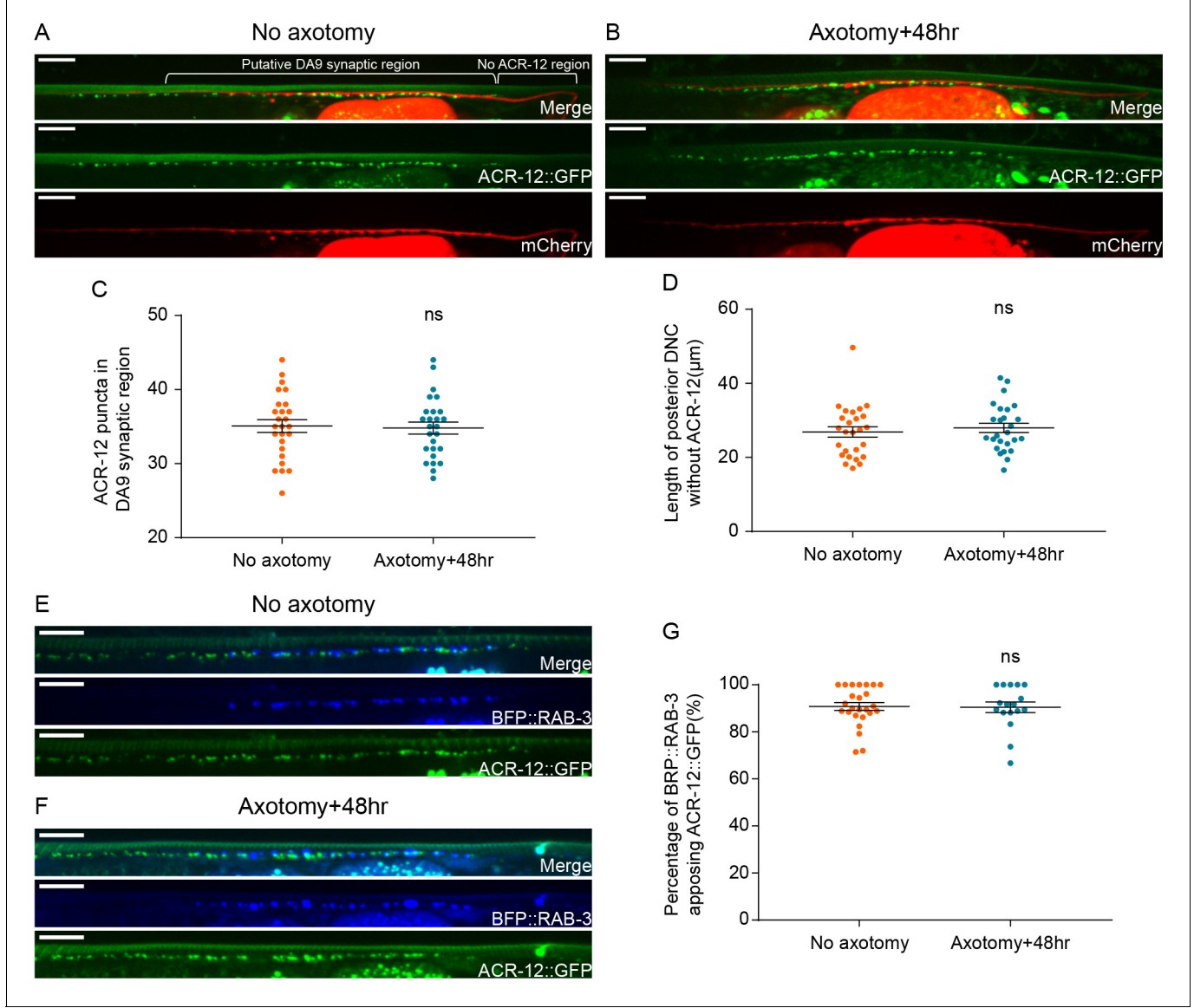

**Figure 3.** Postsynaptic receptors to DA9 maintain their localization and are aligned with DA9 SV clusters after axotomy. (**A**) Labeling of DA9 axon and ACR-12 in GABA motor neurons in intact animals (2d old adults). Scale bars = 10 μm. (**B**) Labeling of DA9 axon and ACR-12 in GABA motor neurons in axotomized animals. Scale bars = 10 μm. (**C**) ACR-12::GFP puncta number in the putative DA9 synaptic region (~85 μm anterior from the most posterior ACR-12 puncta in the dorsal nerve cord) in both intact and axotomized animals. ns, not significant. Unpaired t test. (**D**) Length of posterior DA9 axon without ACR-12 in both intact and axotomized animals. ns, not significant. Unpaired t test. (**E**) BFP::RAB-3 puncta in DA9 axon appose ACR-12::GFP puncta in the postsynaptic GABA motor neurons in intact animals. Scale bars = 10 μm. (**F**) BFP::RAB-3 puncta in DA9 axon appose ACR-12::GFP puncta in the postsynaptic GABA motor neurons 48 hr after axotomy. Scale bars = 10 μm. (**G**) Percentage of BFP::RAB-3 apposing ACR-12::GFP in intact and axotomized animals. ns, not significant. Unpaired t test.

DOI: https://doi.org/10.7554/eLife.38829.007

The following figure supplement is available for figure 3:

**Figure supplement 1.** No change of postsynaptic receptors in body wall muscles after axotomy.

DOI: https://doi.org/10.7554/eLife.38829.008

localization of postsynaptic receptors to the DA9 synaptic region remain constant after injury and regeneration of the DA9 axon.

To determine whether the newly regenerated axonal SV clusters are correctly aligned with the postsynaptic receptors, we co-labeled ACR-12 with GFP in the GABA neurons and RAB-3 with BFP in DA9 (*Chai et al., 2012*; *Petrash et al., 2013*). In intact animals, about 90% of the BFP::RAB-3

puncta apposed or overlapped with ACR-12::GFP puncta (*Figure 3E and G*), consistent with the fact that DA9 normally forms dyadic synapses onto VD GABA neurons (as well as dorsal BWMs). 48 hr after axotomy, 90% of the regenerated BFP::RAB-3 puncta in DA9 axon again apposed the ACR-12: GFP puncta (*Figure 3F and G*). Therefore, the alignment of pre- and postsynaptic sites is restored after axotomy and regeneration to normal levels.

To determine whether the new, ectopic dendritic SV clusters formed after axotomy affect post-synaptic receptors in the ventral body wall muscles, we analyzed the colocalization of ACR-16::GFP puncta in ventral muscle with the DA9 dendrite in the ventral nerve cord. In intact animals, ACR-16 puncta overlapped extensively with the DA9 dendrite (*Figure 3—figure supplement 1A–1A''*). This result is expected, because the DA9 dendrite is bundled with axons of other cholinergic neurons in the ventral nerve cord that synapse onto the ventral BWMs (*Hall and Russell, 1991b*). 48 hr after axotomy, the number of ACR-16 puncta juxtaposed to the DA9 dendrite was similar to uninjured animals (*Figure 3—figure supplement 1B and E*). Thus, the dendritic SV clusters that form in DA9 after axotomy do not induce significant numbers of new postsynaptic receptors in the ventral BWMs.

## Aberrant functional recovery of a single neuron-driven behavior

To determine the effects on function of these morphological changes in DA9 after axotomy, we sought to establish a behavioral assay entirely dependent on DA9 synaptic output. We expressed Chrimson, a red-shifted channel rhodopsin, in DA9 using the DA9-specific *itr-1 pB* promoter and activated it with green light (*Klapoetke et al., 2014*; *Schild and Glauser, 2015*). The anatomy of DA9 predicts that it will activate the posterior dorsal BMWs and inhibit the posterior ventral BMWs, causing the tail to bend dorsally (*Figure 4A*). Indeed, when 5 s of light stimulation was applied, freely behaving Chrimson-expressing animals displayed strong dorsal tail bending (*Figure 4B*). We quantified this behavior by measuring the tail angle using the WormLab software package. Animals that expressed Chrimson and were cultured with all-trans-retinal (ATR) showed an average of 80 degrees of dorsal bending, time-locked to the light stimulus (*Figure 4C*). Dorsal bending in response to light was observed in every animal (N = 42). After the stimulus ended, the tail gradually returned to its normal position. Animals that did not express Chrimson or that were not given ATR, showed no response to light stimulation (*Figure 4—figure supplement 1A, B and D*). In addition, the behavior of Chrimson-expressing animals cultured with ATR was abolished 12 hr after DA9 cell body was ablated by laser (*Figure 4—figure supplement 1C and D*). Thus, this assay allows the precise analysis of the ability of a single neuron to drive behavior, and (in intact animals) has very low levels of inter-animal variability.

We then examined the ability of DA9 to mediate tail bending after acute axon injury. 5 min after axotomy of DA9, light activation of DA9 did not cause dorsal bending behavior (*Figure 4D*). These data confirmed that the bending behavior was specifically driven by DA9, and indicate that axotomy proximal to the known DA9 synaptic region completely disconnects DA9 from its outputs. Even though the distal fragment containing the synaptic puncta contained Chrimson and was exposed to light (together with the rest of the animal), this was insufficient to trigger tail bending. Thus, Chrimson-mediated neuronal output in DA9 requires connection to the cell body.

48 hr after axon injury, when axon and synapse regeneration was complete (*Figure 2E and G*), we found that behavior had partially recovered (*Figure 4G*). At 72 hr, consistent with the lack of further morphological growth after 48 hr, there was no further behavioral recovery (*Figure 4H*). Thus, the maximum behavioral recovery was on average about 1/3 of the control animals (*Figure 4I*, cut 72 hr vs control, mean = 26.3° and 70.5°, p<0.0001, unpaired t test). A major reason for limited recovery on average was that recovered behavior after injury was far more variable than behavior in intact animals: while more than half of injured animals recovered dorsal bending, many did not. Nevertheless, these data indicate that at least some of the newly formed SV puncta in the axon are functional regenerated synapses. To confirm that the behavior recovery was indeed a result of synaptic regeneration of DA9 (rather than some other form of plasticity), we recut DA9 in animals that displayed behavioral recovery at 48 hr, and measured the behavior 12 hr later, before a second round of regeneration could occur. Recovered dorsal bending was completely abolished after recutting, indicating that behavioral recovery was due to DA9 regeneration (*Figure 4J and K*). Together, these experiments indicate that the single-neuron behavioral assay reveals the functional output of DA9 after axon regeneration.

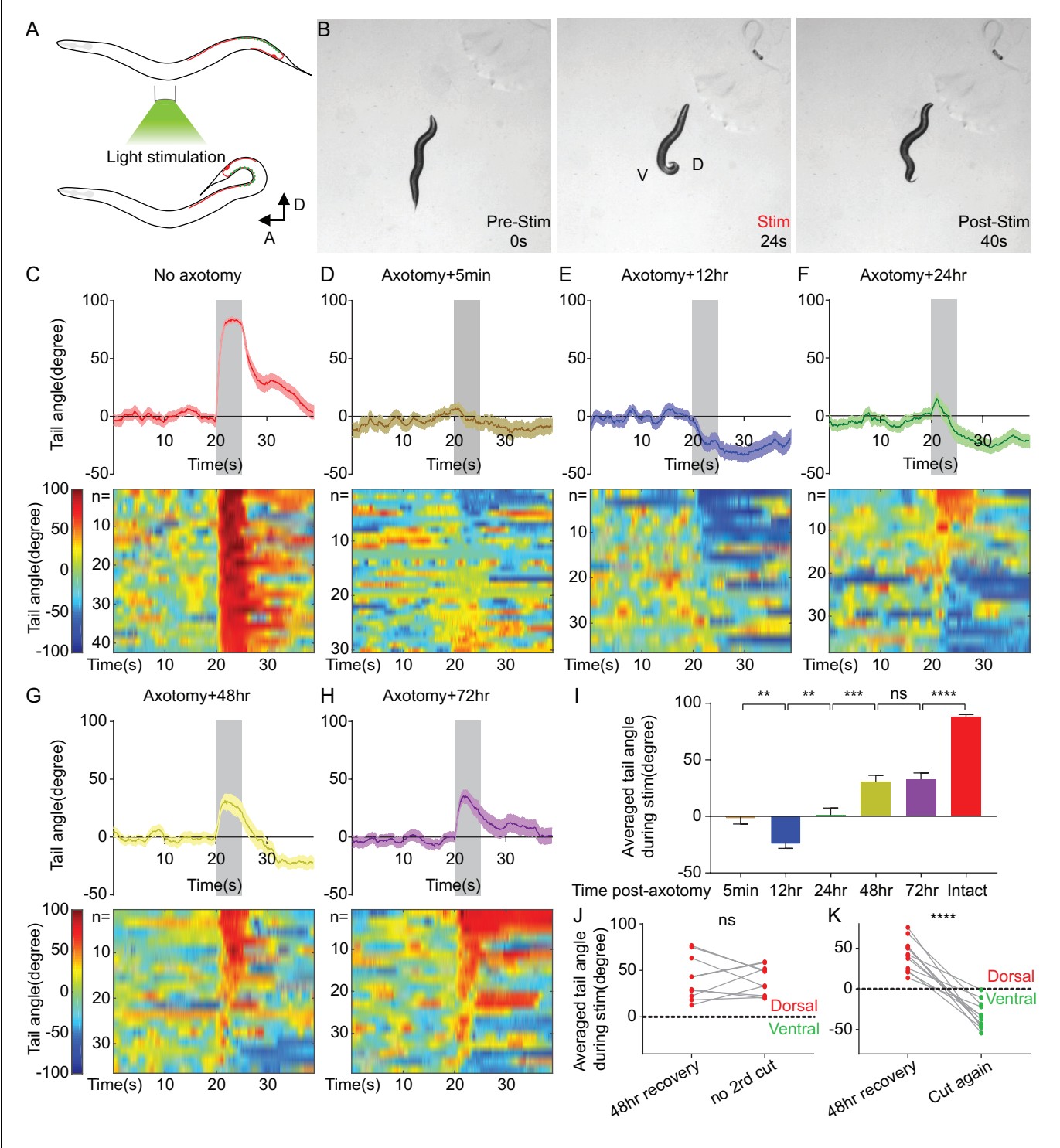

**Figure 4.** Aberrant recovery of a single neuron-driven behavior. (A) Diagram of the optogenetically triggered tail-bending behavior. (B) Images from an example movie showing the dorsal tail-bending behavior. (C–H) Quantification of the tail-bending behavior of intact animals (C), axotomized animals 5 min after axotomy (D), 12 hr after axotomy (E), 24 hr after axotomy (F), 48 hr after axotomy (G) and 72 hr after axotomy (H). Both averaged data (upper, Mean ± SEM) and heat maps (lower) of individual responses are shown. Shaded areas represent the 5 s light stimulation. The color bar indicates the tail angle. Positive numbers indicate dorsal bending, and negative numbers indicate ventral bending. (I) Averaged tail angle during the 5 s stimulation at different timepoints after axotomy of animals in (C–H). Mean and SEM. **p<0.01; ***p<0.001; ****p<0.0001; ns, not significant. Unpaired t test. (J and K) Recovery of dorsal bending is abolished 12 hr after the second cut of the regenerating axon (K) but stays the same in the absence of the second cut (J). ****p<0.0001; ns, not significant. Paired t test.

*Figure 4 continued on next page*

*Figure 4 continued*

DOI: https://doi.org/10.7554/eLife.38829.009

The following figure supplements are available for figure 4:

**Figure supplement 1.** Optogenetically-triggered tail-bending behavior requires Chrimson and ATR and is DA9 specific.
DOI: https://doi.org/10.7554/eLife.38829.010
**Figure supplement 2.** Optogenetically-triggered tail-bending behavior depends on synaptic transmission.
DOI: https://doi.org/10.7554/eLife.38829.011

To determine if the behavioral response is mediated by chemical or electrical synapses, we analyzed the behavior in *unc-13(e51)* animals in which synaptic transmission is blocked (*Richmond et al., 1999*). We detected no behavioral responses in either intact or axotomized animals 48 hr after axotomy (*Figure 4—figure supplement 2*). This suggests that the tail-bending behavior is mediated by chemical synapses in both intact and regenerated animals.

We also analyzed behavior at earlier time points, when SV puncta were at their maximum in the dendrite and axonal regeneration was not yet complete (*Figure 2B*). At 12 hr after axotomy, we found that on average, DA9 activation resulted in ventral rather than dorsal bending (*Figure 4E and I*). Data from individual animals revealed that at 12 hr, more than half of the animals bent their tail ventrally, rather than dorsally. Thus, DA9 activation in these regenerated neurons was causing behavior opposite to uninjured neurons. This aberrant behavior was observed less frequently as regeneration proceeds, but even at 48 and 72 hr after injury some animals still exhibited ventral bending (*Figure 4G and H*). These data demonstrate that although injury and regeneration can restore normal behavior in some individuals, in other individuals injury and regeneration result in formation of a pathological circuit that drives novel and inappropriate behavior.

## Rerouted information transfer in a regenerated circuit

Aberrant ventral bending behavior is maximal at the same time that SV puncta in the dendrite are maximal. This observation suggests a cell-biological explanation for aberrant behavior after regeneration: that the ectopic dendritic SVs are capable of releasing neurotransmitters on the ventral side and activating the ventral BWMs. These muscles normally receive input from other neurons in the ventral nerve cord, and the AChRs of these muscles are normally in close proximity to the DA9 dendrite (*Figure 1A*; *Figure 3—figure supplement 1*). As more new SV puncta are added to the dorsal axon later, release at the dorsal side eventually predominates (*Figure 4E–4H*). To test this hypothesis and to better understand functional regeneration and circuit plasticity, we combined optogenetic stimulation (again using Chrimson expressed in DA9) and $Ca^{2+}$ imaging (using GCaMP6 expressed in body wall muscles) to monitor the activity of postsynaptic BWMs during DA9 regeneration. This approach allowed us to monitor $Ca^{2+}$ levels in both dorsal and ventral BWMs simultaneously during activation of DA9. For this analysis, we focused on $Ca^{2+}$ level changes during DA9 stimulation.

In intact animals, we found that dorsal BWMs displayed robust increases in $Ca^{2+}$ levels during light stimulation of DA9, consistent with DA9 directly activating these muscles (*Figure 5A and C*). We also observed simultaneous decreases in ventral BWM $Ca^{2+}$ levels, presumably mediated by DA9 activation of inhibitory GABAergic VD motor neurons as predicted by the wiring diagram (*Figure 5F and J*). These effects on the muscles should trigger dorsal muscle contraction and ventral relaxation, resulting in the dorsal bending we observed in our behavioral analysis. Importantly, animals cultured without ATR displayed no $Ca^{2+}$ changes at all during stimulation (*Figure 5—figure supplement 1B*), indicating that all the calcium transients are specifically due to activation of Chrimson in DA9. Thus, simultaneous stimulation and calcium imaging allows specific interrogation of functional connectivity in this simple motor circuit.

In sharp contrast to the results in uninjured animals, we found that 12 hr after axotomy, dorsal BWMs showed little $Ca^{2+}$ increase in response to light stimulation (*Figure 5B and E*). The maximum $Ca^{2+}$ amplitude in dorsal BWMs 12 hr after axotomy was about 12% of the control animals (*Figure 5G*, mean = 0.23 and 1.84 with and without axotomy). Further, DA9 activation 12 hr after axotomy drove strong $Ca^{2+}$ increases in ventral BWMs, consistent with the dendritic SV localization and the ventral bending behavior at the 12 hr time point (*Figure 5F and H*). 48 hr after axotomy, DA9 activation was more effective at driving dorsal BWMs, which displayed $Ca^{2+}$ increases about

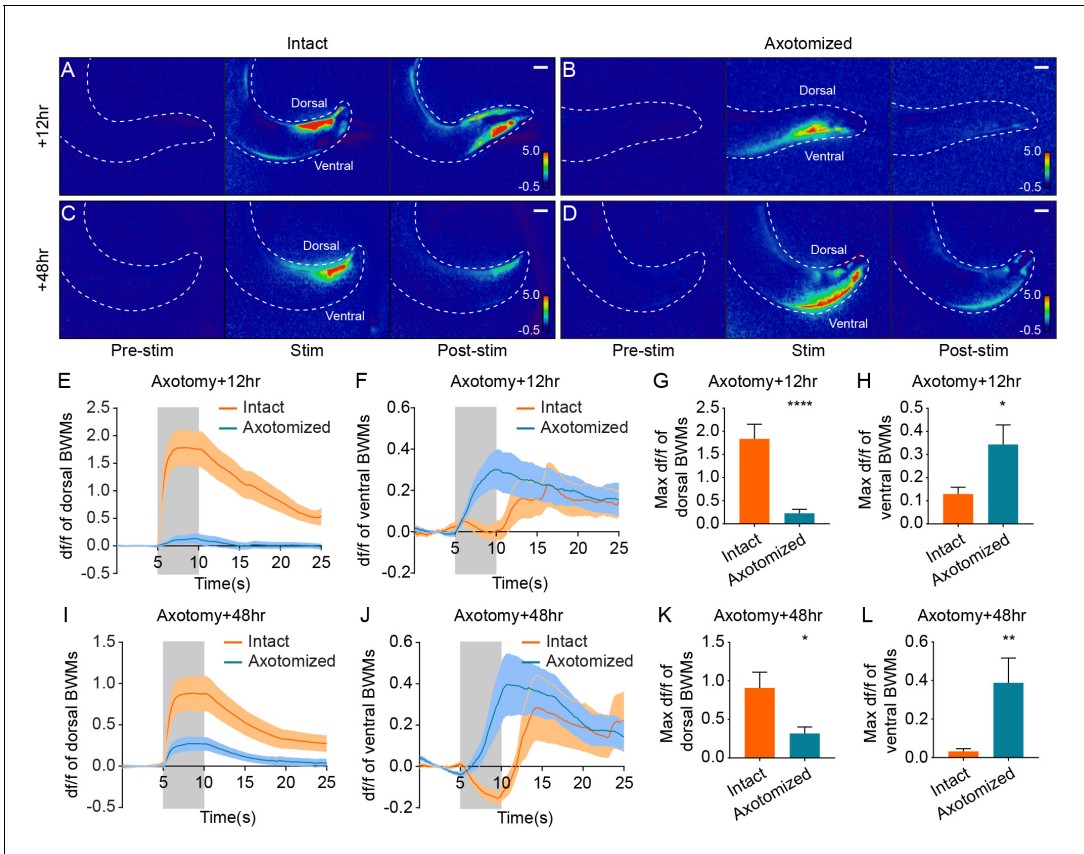

**Figure 5.** Simultaneous optogenetic stimulation and calcium imaging reveals rerouted information transfer in the regenerated circuit. (A–D) Heat maps of calcium signal fold changes (df/f) in intact (A and C) and axotomized animals (B and D) 12 hr or 48 hr after axotomy. White dashed lines delineate the animals' tails. Scale bars = 20 μm. (E–H) Calcium signal traces (E and F) and peak amplitudes during stimulation (G and H) of both dorsal and ventral BWMs in intact and axotomized animals 12 hr after axotomy (n = 29 for axotomized and 25 for intact animals). Shaded areas indicate the 5 s light stimulation. Mean ± SEM. *p<0.05; ****p<0.0001. Unpaired t test. (I–L) Calcium signal traces (I and J) and peak amplitudes during stimulation (K and L) of both dorsal and ventral BWMs in intact and axotomized animals 48 hr after axotomy (n = 30 for axotomized and 31 for intact animals). Shaded areas indicate the 5 s light stimulation. Mean ± SEM. *p<0.05; **p<0.01. Unpaired t test.

DOI: https://doi.org/10.7554/eLife.38829.012

The following figure supplement is available for figure 5:

**Figure supplement 1.** Optogenetically induced muscle calcium response requires all-trans retinal and depends on synaptic transmission.

DOI: https://doi.org/10.7554/eLife.38829.013

35% of the controls, confirming that new synapses in the axon are functional (*Figure 5D, J and K*; mean = 0.32 and 0.91 with and without axotomy in *Figure 5K*). Even at the 48 hr time point, however, the ventral BWMs also displayed $Ca^{2+}$ increases, suggesting that the ectopic synapses in the dendrite are still functional at 48 hr after axotomy (*Figure 5J and L*). Together, these data demonstrate that information transfer—the fundamental role of neurons—is rerouted after axon injury and regeneration.

To determine whether the $Ca^{2+}$ responses are mediated by chemical or electrical synapses, we examined $Ca^{2+}$ signals in the *unc-13(e51)* background. No $Ca^{2+}$ signals were detected in either intact or axotomized animals 48 hr after axotomy (*Figure 5—figure supplement 1C and D*). These data indicate that postsynaptic calcium responses due to DA9 stimulation are mediated by chemical synapses in both intact and regenerated axons, consistent with the results from behavioral analysis (*Figure 4—figure supplement 2*).

## Aberrant information transfer is independent of *dlk-1* and suppresses behavioral recovery

Even when regeneration is complete and axonal synapses have reformed (at 48 and 72 hr), recovery of the ability of DA9 to drive behavior is only partial. We hypothesized that at least part of this deficit is due to aberrant information transfer at these later time points. Specifically, release of SVs from the ectopic synapses in the dendrite would be expected to suppress dorsal bending of the tail, by exciting ventral BWMs and also by exciting DD GABA neurons that would in turn inhibit dorsal BWMs according to the wiring diagram (*Schuske et al., 2004*). We tested the ability of dendritic release to drive behavior at these late time points by using mutants in *dlk-1(ju476)*, a MAPKKK that is a key regulator of regeneration in *C. elegans* and other species (*Hammarlund et al., 2009*; *Shin et al., 2012*; *Xiong et al., 2010*; *Yan et al., 2009*). Consistent with these previous results, we found that DA9 axons completely failed to regenerate in *ric-7(n2657)*; *dlk-1(ju476)* animals (*Figure 6A–6C*). Despite this lack of axonal response, however, *dlk-1(ju476)* mutant animals still mislocalized SV puncta ectopically to the dendrite after axotomy, at similar levels as controls (*Figure 6B and D*). Thus, the *dlk-1(ju476)* background enables analysis of the function of the dendritic SVs in the absence of the SVs in the regenerating axon. Loss of *dlk-1* had no detectable effect on the morphology (*Figure 6A*) or function of DA9; in the absence of injury, *dlk-1(ju476)* mutant animals showed robust dorsal bending of the tail (*Figure 6—figure supplement 1A and C*). However, when DA9 was activated in *dlk-1(ju476)* mutant animals 12 or 48 hr after axotomy, almost all animals bent their tail ventrally (*Figure 6E*, *Figure 6—figure supplement 1B–1D*). Together these data indicate that ectopic synapses in the DA9 dendrite can mediate ventral bending even at 48 hr after axotomy. These data also indicate that mislocalization of SVs to the dendrite in response to injury is independent of *dlk-1* function.

To test the idea that aberrant dendritic SV release suppresses behavioral recovery after axotomy, we eliminated dendritic release by removing the dendrite after axon regeneration was complete. 48 hr after axotomy, we assessed behavior, severed the dendrite, and assessed behavior again 2 hr later. We found that removing the dendrite increased dorsal bending about twofold (*Figure 6F and G*). This result indicates that even after regeneration is complete, misdirected SV release from the dendrite suppresses behavioral recovery.

However, even after dendrite removal, DA9-driven behavior still did not reach the level observed in uninjured controls (*Figure 6G*, 48 hr recovery, recovery +dendrite cut and no cut control, mean = 20.0°, 38.2° and 77.4°, respectively). Thus, axonal activity is only able to drive behavior to about 50% of pre-injury levels even when dendritic suppression is removed. These functional deficits in regenerated synapses are consistent with the molecular deficits we observed in regenerated synapses, where many regenerated synapses have undetectable levels of the active zone components UNC-10 and CLA-1S (*Figure 2—figure supplement 2*). These data suggest that the failure of regeneration to restore particular molecular components to synapses results in persistent functional deficits.

## Dendritic microtubule polarity is maintained after axotomy

We next sought to understand the mechanism of ectopic synapse formation in the dendrite. In some cases, axonal injury to a neuron can cause one of its dendrites to become more axon-like (*Dotti and Banker, 1987*; *Gomis-Rüth et al., 2008*; *Stone et al., 2010*). This conversion can be accompanied by a change in microtubule (MT) polarity (*Stone et al., 2010*; *Takahashi et al., 2007*). MTs in *C. elegans* axons are uniformly polarized with their plus-ends oriented away from the cell body, while MTs in dendrites are largely oriented with minus-ends facing the distal tip (*Maniar et al., 2011*; *Yan et al., 2013*). If axon injury to DA9 results in a change in MT polarity, so that dendritic MTs become axon-like with plus-ends out, this polarity change could account for SV mislocalization to the DA9 dendrite after axotomy. We examined microtubule polarity in DA9 using live imaging of fluorescently tagged MT plus-end-tracking protein EBP-2, the *C. elegans* homolog of human EB1 (*Srayko et al., 2005*; *Yan et al., 2013*) (*Figure 7A–7E*). To examine the effect of injury, imaging was done 12 hr after injury, a time point when dendritic SVs were abundant (*Figure 2H*). We found that in uninjured DA9 neurons, MT polarity in the axon was nearly completely plus-end out, while MT polarity in the dendrite was dominated by plus-end in MTs, consistent with previous results (*Maniar et al., 2011*; *Yan et al., 2013*) (*Figure 7F*). Surprisingly, however, axon injury did not result

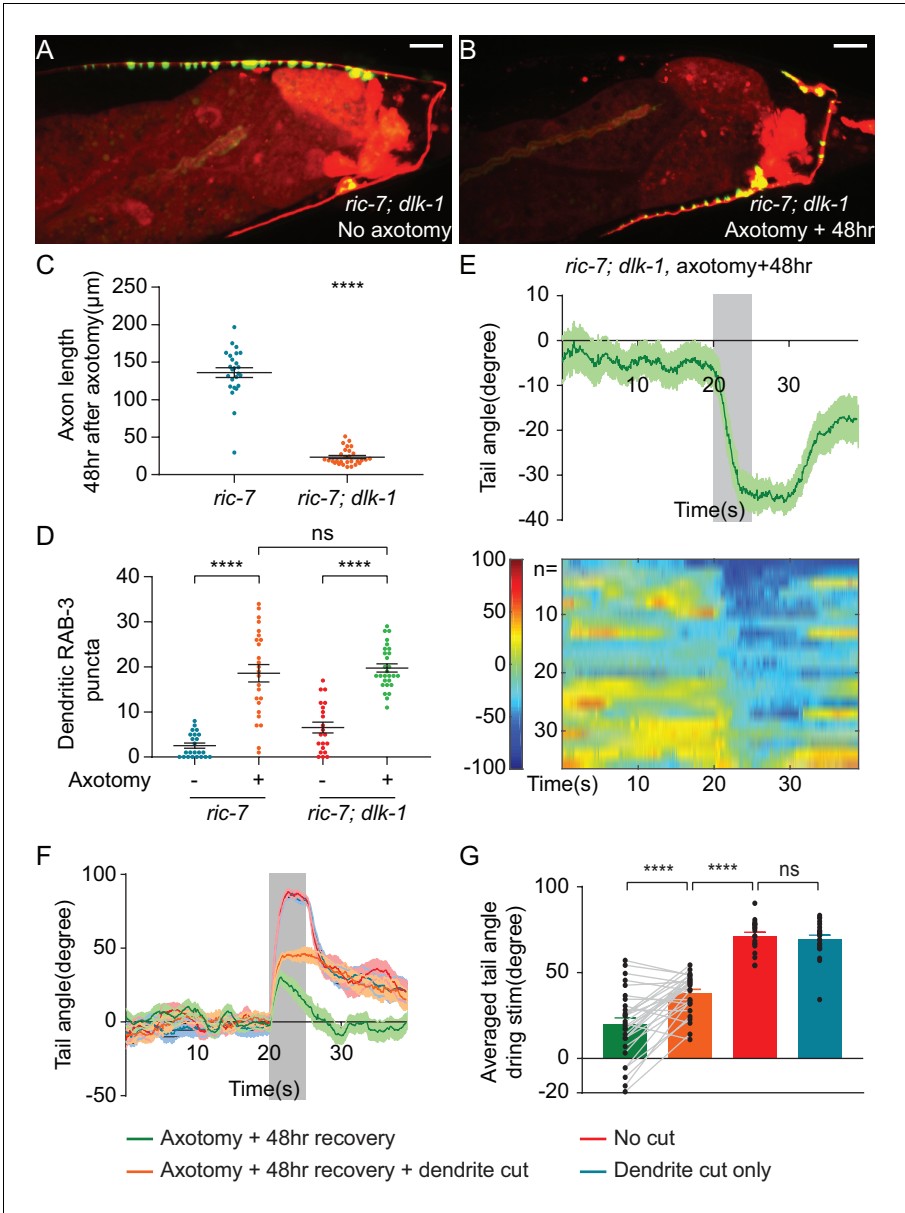

**Figure 6.** Aberrant information transfer is independent of *dlk-1* and suppresses behavioral recovery. (**A and B**) GFP::RAB-3 and mCherry labeling of DA9 in intact and axotomized *ric-7(n2657); dlk-1(ju476)* animals 48 hr after axotomy. Scale bars = 10 μm. (**C and D**) Comparison of length of regenerating axons (**C**) and number of dendritic RAB-3 puncta (**D**) between *ric-7(n2657)* and *ric-7(n2657); dlk-1(ju476)* animals 48 hr after axotomy. Mean ± SEM. ****p<0.0001; ns, not significant. Unpaired t test. (**E**) Ventral tail-bending behavior of *ric-7(n2657); dlk-1(ju476)* animals 48 hr after axotomy. The shaded area indicates the 5 s light stimulation. Mean ± SEM. (**F**) Traces of tail-bending behavior of *ric-7(n2657)* intact (red) and axotomized animals (green) 48 hr after axotomy. Dendrites of axotomized (orange) and intact animals (blue) were then cut and behavior was measured 2 hr later. Mean ± SEM. The shaded area indicates the 5 s light stimulation. (**G**) Averaged tail angles during stimulation of the animals in (**F**). Mean and SEM. ****p<0.0001; ns, not significant. Paired t test between green and orange. Unpaired t test between orange, red and blue.

DOI: https://doi.org/10.7554/eLife.38829.014

The following figure supplement is available for figure 6:

**Figure supplement 1.** Behavior response of *ric-7(n2657); dlk-1(ju476)* animals with and without axotomy.
DOI: https://doi.org/10.7554/eLife.38829.015

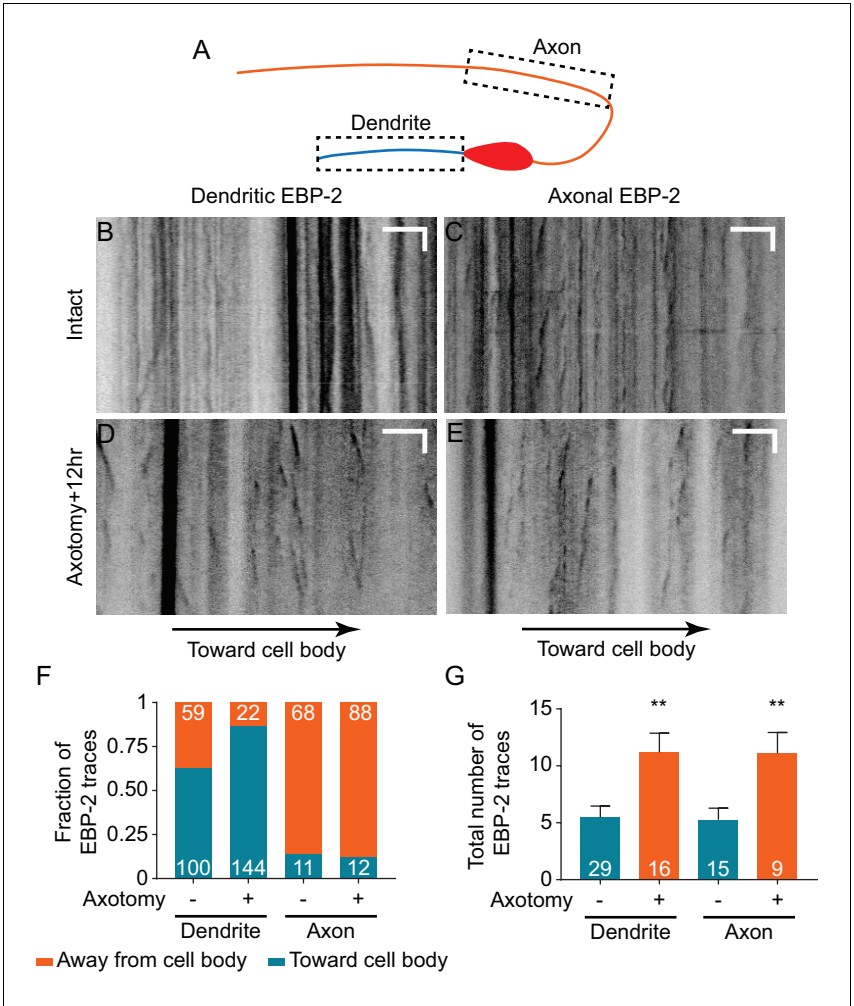

**Figure 7.** Dendritic microtubule polarity is maintained after axotomy. (**A**) Diagram of regions of EBP-2::GFP imaging in DA9. (**B–E**) Kymographs of EBP-2 traces in both DA9 dendrite and axon in intact and axotomized animals 12 hr after axotomy. Scale bars are 5 μm and 5 s. (**F**) Direction of EBP-2 traces in DA9 dendrite and axon. Dendritic microtubule polarity is opposite to axonal microtubule polarity. Numbers represent the number of EBP-2 traces. (**G**) Total number of EBP-2 traces in DA9 dendrite and axon with and without axotomy. Numbers represent the number of animals. Mean and SEM. **$p < 0.01$. Unpaired t test.

DOI: https://doi.org/10.7554/eLife.38829.016

The following figure supplement is available for figure 7:

**Figure supplement 1.** ACR-2, a subunit of the nicotinic acetylcholine receptor, maintains dendritic localization after axotomy.

DOI: https://doi.org/10.7554/eLife.38829.017

in polarity reversal in the axon or dendrite; in fact, in the dendrite after axon injury plus-end in MTs were even more dominant than before injury (*Figure 7F*).

Although axotomy did not reverse the polarity of MTs, it did have a significant effect on MT dynamics in both axons and dendrites. In both compartments, we observed an increased number of EBP-2 traces (*Figure 7G*), which has also been observed in *C. elegans* mechanosensory neurons (*Chuang et al., 2014*; *Ghosh-Roy et al., 2012*) and *Drosophila* sensory neurons (*Stone et al., 2010*). Overall, neuronal injury in DA9 results in increased MT dynamics but no change in MT polarity. Thus, dendritic SV mislocalization is not a consequence of MT polarity changes.

To further test the idea that axon injury might convert the DA9 dendrite into an axon-like process, we examined the localization of the dendritic nicotinic acetylcholine receptor (nAChR) subunit *acr-2* (*Squire et al., 1995*). We found that in uninjured DA9 neurons, ACR-2::GFP was highly enriched in

the dendrite and soma and was absent from the axon (*Figure 7—figure supplement 1A*), as previously described (*Barbagallo et al., 2010*; *Qi et al., 2013*; *Yan et al., 2013*). 12 hr after injury, when ectopic SVs are highly enriched in the dendrite (*Figure 2H*), localization of ACR-2::GFP was not different from uninjured neurons (*Figure 7—figure supplement 1B*). Together, these results indicate that axon injury does not globally push the DA9 dendrite toward becoming axon-like, and suggest that more specific cellular mechanisms must mediate ectopic SV localization.

## Dynein-mediated transport is required for SV mislocalization to the dendrite after injury

Previous studies have shown that UNC-104/KIF1A is the primary motor responsible for transporting SVs towards the MT plus-end (*Hall and Hedgecock, 1991a*; *Okada et al., 1995*; *Pack-Chung et al., 2007*), while the cytoplasmic dynein complex is involved in MT minus-end directed transport in neurons (*Goldstein and Yang, 2000*; *Koushika et al., 2004*). Since MT polarity is maintained after axotomy, so that dendritic MTs are primarily plus-ends in, we reasoned that ectopic synapse formation in the dendrite may be mediated by the minus-end-directed dynein motor complex. Dendritic accumulation of synaptic vesicles and dense-core vesicles has been observed in *cyy-1* and *cdk-5* mutant animals due to alterations in dynein-mediated transport (*Goodwin et al., 2012*; *Ou et al., 2010*). To test this hypothesis, we examined SV puncta in the dendrite in *dhc-1(js319)* mutant animals (*Koushika et al., 2004*). These mutants had largely normal SV distribution in DA9 prior to injury (*Figure 8—figure supplement 1A and B*). However, after injury we found that the *dhc-1(js319)* mutation completely suppressed the dendritic accumulation of SVs at 48 hr after axotomy (*Figure 8B and E*). Partial suppression was obtained in animals mutant for another component of the dynein complex, *nud-2(ok949)*, which is the worm ortholog of Nudel (*Figure 8C and E*). Together these data indicate that dynein-mediated transport is required for SV mislocalization in the dendrite in response to axotomy.

## JNK-1 is required for formation of ectopic synapses and limits behavioral recovery

Since ectopic SV localization to the dendrite is triggered by axon injury, we next sought to find mechanisms that link axonal injury to changes in SV localization. The best-characterized injury-sensing pathway in *C. elegans* is the DLK-1 MAP kinase pathway (*Ghosh-Roy et al., 2010*; *Hammarlund et al., 2009*; *Nix et al., 2011*; *Yan and Jin, 2012*; *Yan et al., 2009*), but SV localization to the dendrite after injury is independent of *dlk-1* (*Figure 6B and D*). We then hypothesized that the JNK3 homolog *jnk-1* might play a role for two reasons (*Kawasaki et al., 1999*). First, *jnk-1* has been shown to be involved in the axon injury response in *C. elegans* GABA neurons. Loss of *jnk-1* results in improved morphological regeneration, while *jnk-1* overexpression inhibits morphological regeneration (*Nix et al., 2014*). Second, although *jnk-1* is not normally required for transport or localization of synaptic proteins in DA9, loss of *jnk-1* largely suppresses the synaptic phenotypes of *arl-8* mutants, suggesting that in some contexts *jnk-1* can regulate SV localization (*Wu et al., 2013*). We therefore examined whether *jnk-1* is involved in ectopic dendritic synapse formation in DA9 after axotomy. In uninjured neurons, synapse location was largely unaffected by loss of *jnk-1*, with SV puncta confined to a specific region of the axon, and essentially absent from the dendrite (*Figure 8—figure supplement 1C and E*), consistent with previous results (*Wu et al., 2013*). By contrast, after axon injury *jnk-1(gk7)* mutant animals showed very few SV puncta in the dendrite 48 hr (*Figure 8D and E*) and 12 hr after axotomy (*Figure 8—figure supplement 1F and G*). Thus, ectopic synapse formation in the dendrite after axon injury requires the JNK3 homolog *jnk-1*.

Ectopic synapses limit behavioral recovery after nerve injury. Therefore, we tested the idea that *jnk-1* mutant animals might have improved behavioral recovery. Without axon injury, *jnk-1(gk7)* mutants showed robust dorsal bending in response to light stimulation, although the degree of dorsal bending was smaller compared to controls (*Figure 8F and G*, *jnk-1(gk7); ric-7(n2657)* intact vs *ric-7(n2657)* intact, mean = 59.7° and 70.1°, p=0.0003, unpaired t test). By contrast, 48 hr after axotomy, the *jnk-1(gk7)* mutants showed improved dorsal bending behavior compared to controls (*Figure 8F and G*), similar to the improved behavior seen after dendrite removal (*Figure 6F and G*). Thus, altering neuronal cellular mechanisms to correctly direct information transfer in regenerated circuits improves behavioral recovery after nerve injury.

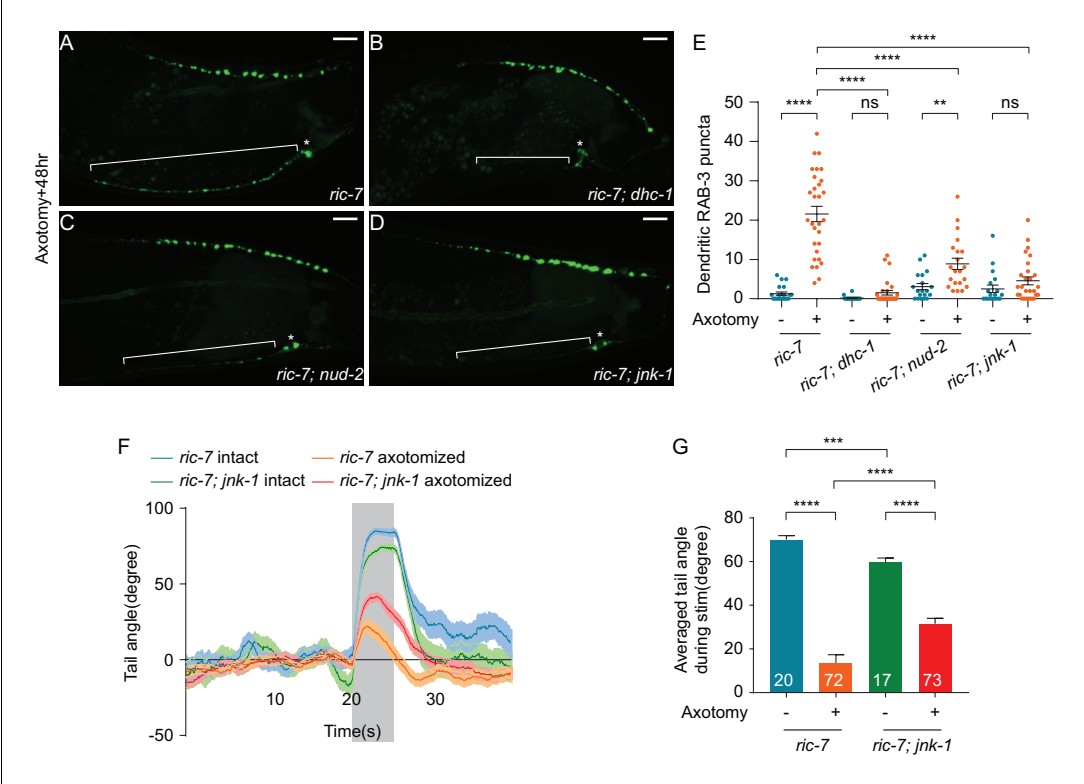

**Figure 8.** JNK-1 and dynein-mediated transport mediate SV mislocalization to the dendrite and loss of JNK-1 improves behavioral recovery. (A–D) GFP::RAB-3 localization 48 hr after axotomy in *ric-7(n2657)* (A), *ric-7(n2657); dhc-1(js319)* (B), *ric-7(n2657); nud-2(ok949)* (C) and *ric-7(n2657); jnk-1(gk7)* (D) animals. Asterisk indicates cell body and bracket indicates dendrite. Scale bars = 10 μm. (E) Quantification of the number of dendritic RAB-3 puncta in intact and axotomized animals 48 hr after axotomy. Mean ± SEM. **p<0.01; ****p<0.0001; ns, not significant. Unpaired t test. (F) Traces of the tail-bending behavior of *ric-7(n2657)* and *ric-7(n2657); jnk-1(gk7)* animals with and without axotomy 48 hr after axotomy. The shaded area indicates the 5 s stimulation. Mean ± SEM. (G) Averaged tail angle during stimulation of the animals in (F). Numbers represent the number of animals. Mean and SEM. ***p<0.001; ****p<0.0001. Unpaired t test.

DOI: https://doi.org/10.7554/eLife.38829.018

The following figure supplement is available for figure 8:

**Figure supplement 1.** DA9 in *dhc-1(js319)* animals develops normally and loss of JNK-1 rescues RAB-3 mis-localization in DA9 dendrite 12 hr after axotomy.

DOI: https://doi.org/10.7554/eLife.38829.019

## Discussion

### Analyzing functional regeneration in DA9

This study establishes a new system for the study of axon regeneration in the DA9 neuron of *C. elegans.* We find that severing DA9 close to the dorsal nerve cord allows regeneration across the former synaptic area to occur in essentially all animals studied. Further, by using the *ric-7(n2657)* genetic background to promote removal of the distal axon fragment, regenerative growth in DA9 occurs without confounding issues of axonal fusion or other potential effects of the remaining fragment. Previous studies in *C. elegans* have shown that the regenerating mechanosensory PLM neuron can reconnect to its distal axon segment after axotomy. This fusion event essentially restores the neuron back to its pre-injury state, without the need to rebuild synapses, and restores the full function of the circuit as measured by a light touch assay (*Abay et al., 2017*; *Neumann et al., 2011*). In this work, by using the *ric-7(n2657)* background to promote distal segment removal, functional recovery can occur only if new, functional synapses are rebuilt onto relevant targets. In combination, our approach in DA9 in the *ric-7(n2657)* background allows for the first time detailed morphological analysis of synapse reformation in a single neuron during axon regeneration. Further, by applying the modern neuroscience tools of calcium imaging and optogenetics to DA9 and its target muscles,

it is possible to analyze the function of this single neuron in its endogenous circuit and also with respect to its ability to drive behavior. We expect these approaches to be useful for studies relating synapse formation to circuit function and behavior, both in the context of axon regeneration and also in uninjured neurons.

A potential concern is that loss of *ric-7(n2657)* may cause phenotypes that interact with the question under study. *ric-7(n2657)* encodes a protein of approximately 700 residues, with no clear vertebrate orthologs or functional domains. *ric-7(n2657)* was first identified in a screen for genes involved in neuropeptide secretion. In *ric-7(n2657)* mutants, neuropeptide secretion is reduced while acetylcholine release and response are unaltered (*Hao et al., 2012*). *ric-7(n2657)* mutants were also found to have greatly enhanced degeneration of axon segments after axotomy, and this phenotype was found to be due to reduced localization of mitochondria to axons (*Rawson et al., 2014*). Interestingly, although mitochondria function is important for regeneration in *C. elegans* GABA neurons and in mammalian models (*Cartoni et al., 2016*; *Han et al., 2016*; *Zhou et al., 2016*), we found that *ric-7(n2657)* animals have improved axon regeneration (*Figure 1—figure supplement 1E–1H*). Thus, in DA9 the loss of *ric-7* increases degeneration without negatively impacting regeneration. Further, we found that synaptic vesicle localization to the dendrite in response to injury is not dependent on *ric-7(n2657)*, as it occurs at the same level in animals that are wild type and mutant at this locus (*Figure 2H*). Although the use of *ric-7(n2657)* demands caution, our data indicate that it is a useful and effective tool for studying axon regeneration in DA9.

## Aberrant information rerouting limits behavior recovery after regeneration

We show that DA9 axons do regenerate and form functional synapses in the correct location after axotomy. However, we find that synapses also form ectopically in the dendrite, resulting in aberrant information routing and limiting behavioral recovery. Although ectopic synapse formation is reminiscent of the transformation of a dendrite to an axon after axon removal (*Cho and So, 1992*; *Dotti and Banker, 1987*; *Fenrich et al., 2007*; *Gomis-Rüth et al., 2008*; *Hall et al., 1989*; *Hoang et al., 2005*; *Stone et al., 2010*), we found that it is a fundamentally different process. First, dendrite-axon transformation requires axon removal, while in our experiments the axon is severed distant from the cell body, regenerates, and reforms functional synapses. Second, dendrite-axon transformation is accompanied by a rearrangement of MT polarity from dendritic orientation to axonal orientation (*Stone et al., 2010*; *Takahashi et al., 2007*). In the case of DA9, however, dendritic microtubule polarity as well as nAChR localization is maintained after axotomy (*Figure 7F*, *Figure 7—figure supplement 1*). Thus, our results define a novel cell-biological response to nerve injury.

The ability of dendritic synapses in DA9 to transfer information and cause specific behavioral defects likely depends on the circuit context within which DA9 is embedded. The dendrite of DA9 is in the ventral nerve cord, where it receives inputs from the AVA and AVD command neurons (*Hall and Russell, 1991b*; *White et al., 1986*). What postsynaptic cells might respond to acetylcholine release from the DA9 dendrite to trigger ventral bending? The VNC contains a variety of ACh receptor fields. In particular, ventral muscles make synapses with ventral motor neurons in the VNC, such as VA and VB cells (*White et al., 1986*). Thus, the DA9 dendrite and ventral muscle receptor fields are both localized in the VNC (*Figure 3—figure supplement 1*). Our calcium imaging data indicate that DA9 activation can trigger ventral muscles (*Figure 5B and D*), consistent with the idea that these muscles might respond directly to ACh release from DA9. However, it is also possible that information transfer from DA9 to ventral muscles is not monosynaptic. VNC motor neurons like VA and VB also have cholinergic receptor fields in the VNC and could transfer excitatory signals from the DA9 dendrite to muscle. Thus, the particular anatomy of the *C. elegans* VNC helps determine the effect of dendritic release from DA9.

Synapses are highly organized junctions between pre- and postsynaptic cells, with vesicle release sites in tight apposition to postsynaptic receptors. In both DA9 axon and dendrite, postsynaptic cells may respond to synapse formation after DA9 injury. A key question for future study is to identify the direct synaptic targets of DA9 activity in axon and dendrite, and examine postsynaptic organization in these cells after injury. We found that the new synapses after regeneration in the DA9 axon again were apposed to the original postsynaptic receptors (*Figure 3E–3G*). This result is reminiscent of key findings at the frog neuromuscular junction, in which denervation and reinnervation of muscle resulted in synapse formation at the original locations (*Letinsky et al., 1976*; *Marshall et al., 1977*;

*Rotshenker and McMahan, 1976*). Thus, at least for DA9 axonal synapse regeneration, an attractive hypothesis is that agrin or other components of the extracellular matrix direct the location of synapses during regeneration, ensuring that they are correctly apposed to their postsynaptic partners.

### The role of dynein and JNK-1 in ectopic synapse formation

JNK, or c-Jun N-terminal kinase, is a MAP kinase that has been implicated in a variety of cellular processes such as immunity, stress response, tumor development and apoptosis (*Davis, 2000*). In both *Drosophila* and mouse, JNK signaling is required for efficient axon regeneration (*Ayaz et al., 2008*; *Raivich et al., 2004*). *C. elegans* has multiple JNK orthologs, including *kgb-1* and *jnk-1*. As in *Drosophila* and mouse, *kgb-1* is required for axon regeneration (*Nix et al., 2011*). By contrast, *jnk-1* suppresses axon regeneration, suggesting that different JNK pathways play different roles during regeneration (*Nix et al., 2014*). Here, we found that JNK-1 is required for ectopic synapse formation in the dendrite after axotomy (*Figure 8D*). JNK-1 also limits behavioral recovery (*Figure 8F and G*). Like *jnk-1*, mutants in the dynein complex also rescued ectopic synapse formation (*Figure 8B*). Thus, JNK-1 may modulate dynein-mediated SV transport to promote the formation of ectopic synapses. However, further work is required to determine the cell-biological mechanisms that link *jnk-1* function to the formation of ectopic synapses.

### New synapses in the regenerating axon are not as efficient as intact ones

In our single-neuron system, although the regenerated synapses in the axon are able to drive some behavioral recovery, the level of dorsal tail bending is significantly lower than in intact controls. Even after removing the dendrite, which inhibits dorsal bending, full behavioral recovery is still not achieved (*Figure 6F and G*). To determine the causes for the deficient functional recovery, we characterized the synapses in both intact and regenerated DA9. We demonstrated that DA9 can efficiently regenerate axonal SV clusters, re-establishing both number and localization close to pre-axotomy levels. Postsynaptic receptors are also similar after regeneration, and the regenerated SV clusters are correctly aligned with the postsynaptic sites. However, the active zone proteins UNC-10 and CLA-1S only regenerate partially (*Figure 2—figure supplement 2*), suggesting that some SV clusters represent defective synapses. Since UNC-10 and CLA-1S contribute to synaptic transmission (*Koushika et al., 2001*; *Xuan et al., 2017*), deficits in these molecules or other synaptic components may account for the limits in functional recovery. A more thorough molecular characterization of the regenerated synapses is needed to understand the complete differences between regenerated and intact synapses. Our study highlights the difficulty in restoring normal circuit function after nerve injury, and provides insight into specific cellular choke points that need to be resolved to achieve complete recovery.

## Materials and methods

### Strains

Animals were maintained at 20°C on NGM plates seeded with OP50 *E. coli* according to standard methods. The following strains were obtained from the *Caenorhabditis* Genetic Center (CGC): MT6924 [*ric-7(n2657)*], NM1489 [*dhc-1(js319); jsIs37*], VC8 [*jnk-1(gk7)*], RB1022 [*nud-2(ok949)*], CZ5730 [*dlk-1(ju476)*] and MT7929 [*unc-13(e51)*]. TV12772 [*wyIs386(Pitr-1::acr-2::gfp)*] and IZ556 [*acr-12(ok367); ufIs92(Punc-47::acr-12::gfp_{ICL})*] was provided by the Yogev lab.

The following transgenes were generated by microinjection (Promega 1 kb DNA ladder was used as a filler in the injection mixes): wpIs101 [*Pitr-1 pB::gfp::rab-3::SL2::mcherry*@50 ng/μL; *Pmyo-2::mcherry*@2 ng/μL], wpIs102 [*Pitr-1 pB::gfp::rab-3::SL2::mcherry*@50 ng/μL; *Pmyo-2::mcherry*@2 ng/μL], wpIs109 [*Pitr-1 pB::mcherry*@50 ng/μL; *Podr-1::rfp*@30 ng/μL], wpIs98 [*Pitr-1 pB::gfp::Chrimson::SL2::mcherry*@20 ng/μL; *Podr-1::rfp*@30 ng/μL], wpIs103 [*Pmyo-3::GCaMP6*@80 ng/μL; *Pmyo-2::mcherry*@2 ng/μL], wpEx294 [*Pitr-1 pB::ebp-2::gfp::SL2::mcherry*@50 ng/μL; *Pmyo-2::mcherry*@2 ng/μL], wpEx295 [*Pitr-1 pB::mcherry*@25 ng/μL; *Pmyo-2::mcherry*@2 g/μL], wpEx312 [*Pitr-1 pB::unc-10::gfp::SL2::mcherry*@50 ng/μL; *Pmyo-2::mcherry*@2 ng/μL], wpEx313 [*Pitr-1 pB::cla-1s::gfp::SL2::mcherry*@50 ng/μL; *Pmyo-2::mcherry*@2 ng/μL], wpEx314 [*Pmyo-3::acr-16::gfp*@50 ng/μL; *Pmyo-2::mcherry*@2 ng/μL], wpEx315 [*Pitr-1 pB::bfp::rab-3*@50 ng/μL; *Pmyo-2::mcherry*@2 ng/μL].

## List of strains used in this study

| Strain | Genotype |
| --- | --- |
| XE1967 | wpIs101 [Pitr-1 pB::gfp::rab-3::SL2::mcherry; Pmyo-2::mcherry] |
| XE1931 | wpIs101 [Pitr-1 pB::gfp::rab-3::SL2::mcherry; Pmyo-2::mcherry]; ric-7(n2657) |
| XE1932 | wpIs102 [Pitr-1 pB::gfp::rab-3::SL2::mcherry; Pmyo-2::mcherry]; ric-7(n2657); dlk-1(ju476) |
| XE1990 | wpIs101 [Pitr-1 pB::gfp::rab-3::SL2::mcherry; Pmyo-2::mcherry]; ric-7(n2657); dhc-1(js319) |
| XE1991 | wpIs101 [Pitr-1 pB::gfp::rab-3::SL2::mcherry; Pmyo-2::mcherry]; ric-7(n2657); nud-2(ok949) |
| XE1992 | wpIs101 [Pitr-1 pB::gfp::rab-3::SL2::mcherry; Pmyo-2::mcherry]; ric-7(n2657); jnk-1(gk7) |
| XE1654 | juIs76 [Punc-25::GFP] |
| XE1993 | juIs76 [Punc-25::GFP]; ric-7(n2657) |
| XE1935 | wpIs98 [Pitr-1 pB::gfp::Chrimson::SL2::mcherry; Podr-1::rfp] |
| XE1936 | wpIs98 [Pitr-1 pB::gfp::Chrimson::SL2::mcherry; Podr-1::rfp]; ric-7(n2657) |
| XE1937 | wpIs98 [Pitr-1 pB::gfp::Chrimson::SL2::mcherry; Podr-1::rfp]; ric-7(n2657); dlk-1(ju476) |
| XE1994 | wpIs98 [Pitr-1 pB::gfp::Chrimson::SL2::mcherry; Podr-1::rfp]; ric-7(n2657); jnk-1(gk7) |
| XE1995 | wpIs98 [Pitr-1 pB::gfp::Chrimson::SL2::mcherry; Podr-1::rfp]; ric-7(n2657); wpIs103 [Pmyo-3::GCaMP6; Pmyo-2::mcherry] |
| XE1964 | wpEx294 [Pitr-1 pB::ebp-2::gfp::SL2::mcherry; Pmyo-2::mcherry]; ric-7(n2657) |
| XE1996 | wpEx295 [Pitr-1 pB::mcherry; Pmyo-2::mcherry]; ric-7(n2657); wyIs386(Pitr-1::acr-2::gfp) |
| XE2039 | wpEx312 [Pitr-1 pB::unc-10::gfp::SL2::mcherry; Pmyo-2::mcherry]; ric-7(n2657) |
| XE2040 | wpEx313 [Pitr-1 pB::cla-1s::gfp::SL2::mcherry; Pmyo-2::mcherry]; ric-7(n2657) |
| XE2041 | wpEx314 [Pmyo-3::acr-16::gfp; Pmyo-2::mcherry]; wpIs109 [Pitr-1 pB::mcherry; Podr-1::rfp]; ric-7(n2657) |
| XE2042 | wpEx315 [Pitr-1 pB::bfp::rab-3; Pmyo-2::mcherry]; wpIs109 [Pitr-1 pB::mcherry; Podr-1::rfp]; ufIs92(Punc-47::acr-12::gfp$_{ICL}$)]; ric-7(n2657) |
| XE2043 | wpIs98 [Pitr-1 pB::gfp::Chrimson::SL2::mcherry; Podr-1::rfp]; ric-7(n2657); unc-13(e51) |
| XE2044 | wpIs98 [Pitr-1 pB::gfp::Chrimson::SL2::mcherry; Podr-1::rfp]; wpIs103 [Pmyo-3::GCaMP6; Pmyo-2::mcherry]; ric-7(n2657); unc-13(e51) |
| XE2045 | wpIs109 [Pitr-1 pB::mcherry; Podr-1::rfp]; ufIs92(Punc-47::acr-12::gfp$_{ICL}$)]; ric-7(n2657) |

## Cloning and constructs

Plasmids were assembled using Gateway recombination (Invitrogen). Entry clones were generated by Gibson Assembly when needed.

## Laser axotomy

Laser axotomy was performed as described previously (*Byrne et al., 2011*). In brief, L4 stage animals were immobilized with 0.1 µm diameter polystyrene beads (Polysciences) and mounted on a 3% agarose pad on a glass slide. The animals were visualized with a Nikon Eclipse 80i microscope using a 100x Plan Apo VC lens (1.4 NA). DA9 axons were cut at the posterior part of the asynaptic region using a 435 nm Micropoint laser by 10 pulses at 20 Hz. In some experiments, DA9 dendrites were cut near the cell body or the DA9 cell body was abated. For the axotomy of GABA motor neurons, the first, fourth and seventh axons from the tail were cut at the midpoint of the commissures. Animals were then recovered to OP50 seeded NGM plates and analyzed later.

## Fluorescence microscopy and quantification

To assess DA9 axonal and synaptic regeneration, animals were imaged at different time points after axotomy. Animals were immobilized with 4 – 40 mM levamisole and mounted on a 3% agarose pad on a slide. Images were then acquired as 0.3 µm z stacks on a spinning-disc confocal microscope (PerkinElmer UltraVIEW VoX, Nikon Ti-E Eclipse inverted scope, Hamamatsu C9100-50 camera) with a CFI Plan Fluor 40X oil objective (1.3 NA) using Volocity software (PerkinElmer). Throughout this

study, all images were acquired with the same exposure time, camera sensitivity and laser power. Images were then exported as tiff files and analyzed in ImageJ.

Maximum intensity projection was created from the tiff files for analysis. DA9 axon or dendrite length was measured using an ImageJ plugin, Simple Neurite Tracer (v3.1.0). Injury-induced degeneration was quantified manually with beadings, breaks and clearance of the distal axon fragment as the criteria. To count synaptic vesicle puncta in the DA9 axon, an ROI around the axon was drawn. A default threshold was determined by ImageJ based on the intensity histogram and applied to the ROI. This created a binary mask of SV puncta in the axon. The mask was then processed with the Watershed function in order to separate puncta close to each other. Finally, the number of puncta was measured using the Analyze Particle function in ImageJ. The results were further confirmed manually. This thresholding formula provided a compromise between including all puncta and avoiding background noise.

A different method was used to count SV puncta in the dendrite because in uncut animals the SV signal was extremely weak, so automatic thresholding created artifacts. Briefly, line scanning was performed along the dendrite, and a uniform threshold of 2500 fluorescent intensity was applied. Peaks above the threshold were counted as SV puncta.

The apposition between BFP::RAB-3 and ACR-12::GFP was determined by quantifying the number of BFP puncta that were either apposing or overlapping with GFP puncta.

To measure axon regeneration of GABA motor neurons, animals were immobilized with polystyrene beads on a 3% agarose pad and imaged at 24 hr after axotomy. Images were acquired using a UPLFLN 40X oil objective (1.3 NA) on an Olympus BX61 microscope equipped with an Olympus DSU and a Hamamatsu ORCA-Flash4.0 LT camera. Acquisition was controlled using the MicroManager software. Images were analyzed in ImageJ. Briefly, GABA axon length along the dorsal-ventral body axis was measured and normalized to the full axis length. The distribution of the normalized length was used to compare regeneration.

## Optogenetics and behavior

Animals expressing Chrimson in DA9 were cultured with OP50 containing 10 mM all-trans-retinal (ATR) on NGM plates as previously described (*Schild and Glauser, 2015*). Prior to the behavior assay, animals were transferred to a fresh plate without OP50. The plate was then placed under a Leica M165FC stereo scope. Movies were taken with a Basler acA2440 camera controlled by the WormLab software. A Prior Lumen 200 fluorescent source and a Leica mCherry filter set were used to give light stimulation. Light intensity was about 300 W/m$^2$ throughout the study. Bright field illumination was kept at a low intensity to reduce non-specific activation of Chrimson. The tail bending behavior was analyzed using the Wormlab software. In brief, the animal was detected by thresholding, and then the body midline was segmented evenly with 14 points. The angle between the last three points was used as the tail angle.

## Calcium imaging

Animals expressing Chrimson in DA9 and GCaMP6 in BWMs were cultured with OP50 and ATR. Animals were immobilized by polystyrene beads and mounted on 3% agarose pads on slides. Movies were taken on an Olympus BX61 microscope with a Hamamatsu ORCA-Flash4.0 LT camera. An X-Cite XLED1 was used as the light source (EXCETILAS). Specifically, a blue LED module (BDX (450 – 495 nm)) was used to image GCaMP6 and a red LED module (RLX(615 – 655 nm)) was used to give continuous stimulation. To reduce the activation of Chrimson by blue light, a ND4.0 filter was mounted in front of the BDX module and the power of blue light was set to 10%. Images were acquired at 5 Hz using the HCImageLive software. Movies that had a stable baseline before stimulation were used for analysis in ImageJ. Briefly, an ROI was selected far away from the animal to generate a background fluorescence intensity, $F_b$. ROIs of the posterior dorsal and ventral BWMs were also drawn to detect Ca$^{2+}$ signals, $F_t$. $F_t'=F_t$ $F_b$ was used to calculate Ca$^{2+}$ changes. Then the baseline, $F_0$, was calculated as averaged $F_t'$ from the first 5 s of the movie. Finally, F=($F_t'$-$F_0$)/$F_0$ was used to display Ca$^{2+}$ signal changes.

## Time-lapse imaging and quantification

Time-lapse imaging was performed to visualize EBP-2::GFP traces in DA9 as previously described (*Chen et al., 2011*; *Ghosh-Roy et al., 2012*; *Yan et al., 2013*). In summary, animals were immobilized by 4 – 40 mM levamisole and mounted on 3% agarose pads on slides. Movies were taken on an Olympus BX61 microscope with a Hamamatsu ORCA-Flash4.0 LT camera at 5 or 10 Hz for 1 min. Movies were then analyzed in ImageJ. Briefly, a line was drawn along the dendrite or the axon and the kymograph was generated using the Reslice function. EBP-2 traces were analyzed from the kymograph and confirmed manually in the movie.

## Acknowledgements

We thank the international *C elegans* Gene Knockout Consortium, the *Caenorhabditis* Genetic Center and the Yogev lab for strains. We thank the Colón-Ramos lab for sharing the Chrimson and GCaMP6 plasmids. We thank everyone in the Hammarlund lab for feedback and suggestions.

## Additional information

### Funding

| Funder | Grant reference number | Author |
| --- | --- | --- |
| National Institutes of Health | R01NS094219 | Marc Hammarlund |
| National Institutes of Health | R01NS098817 | Marc Hammarlund |
| China Scholarship Council | CSC-Yale World Scholars in the Biomedical Sciences | Chen Ding |

The funders had no role in study design, data collection and interpretation, or the decision to submit the work for publication.

### Author contributions

Chen Ding, Conceptualization, Formal analysis, Investigation, Visualization, Methodology, Writing—original draft, Writing—review and editing; Marc Hammarlund, Conceptualization, Supervision, Funding acquisition, Methodology, Writing—review and editing

### Author ORCIDs

Marc Hammarlund (iD) http://orcid.org/0000-0002-3068-068X

### Decision letter and Author response

Decision letter https://doi.org/10.7554/eLife.38829.022
Author response https://doi.org/10.7554/eLife.38829.023

## Additional files

### Supplementary files

• Transparent reporting form
DOI: https://doi.org/10.7554/eLife.38829.021

### Data availability

All data generated or analysed during this study are included in the manuscript and supporting files.

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
