## [Decision Letter]

Thank you for submitting your article "Aberrant Information Transfer Interferes with Functional Axon Regeneration" for consideration by *eLife*. Your article has been reviewed by two peer reviewers, and the evaluation has been overseen by a Reviewing Editor and Eve Marder as the Senior Editor. The following individual involved in review of your submission has agreed to reveal his identity: Joshua M Kaplan (Reviewer #2).

The reviewers have discussed the reviews with one another and the Reviewing Editor has drafted this decision to help you prepare a revised submission.

Essential revisions:

1) Validating the synapses:

A) The data on reinnervation from regenerating DA9 seem only based on counting GFP::RAB-3 cluster number. To conclude that these puncta are synapses, the authors should consider analyzing other pre-synaptic (UNC-13, UNC-10, UNC-2) and post-synaptic (ACR-16 and UNC-29) markers. Analyzing new synaptic markers may strengthen the authors' conclusion that the regenerated synapses are functionally weaker than those in unoperated controls (i.e. if regenerated synapses are deficient for key presynaptic components).

B) How are such SV clusters positioned relative to postsynaptic muscle receptor clusters? Do the muscle receptors stay in the same position during the entire period of axon degeneration and regrowth? Could some of the partial or aberrant behavior recovery be due to mis-alignment of new pre- and post-synaptic sites?

C) Likewise, are the 'dendritic SV clusters' inducing new postsynaptic receptors, or altering existing postsynaptic sites? They showed ACR-2::GFP cluster was not affected by injury to axons. How are the injury-induced dendritic RAB-3 clusters formed in relation to ACR-2::GFP?

2) Whether the tail-bending is mediated by chemical or electrical synapses:

Is behavioral recovery and DA9 evoked muscle depolarization mediated by the regenerated synaptic puncta? To support this conclusion, the authors should determine if DA9 mediated behavior and muscle activation (GCAMP) following axotomy are blocked by mutations that block synaptic transmission (e.g. *unc-13, unc-10*, and *unc-2*).

---

## [Author Response]

Essential revisions:1) Validating the synapses:A) The data on reinnervation from regenerating DA9 seem only based on counting GFP::RAB-3 cluster number. To conclude that these puncta are synapses, the authors should consider analyzing other pre-synaptic (UNC-13, UNC-10, UNC-2) and post-synaptic (ACR-16 and UNC-29) markers. Analyzing new synaptic markers may strengthen the authors' conclusion that the regenerated synapses are functionally weaker than those in unoperated controls (i.e. if regenerated synapses are deficient for key presynaptic components).

We analyzed two other pre-synaptic markers: UNC-10, the *C. elegans* homolog of Rim1 (Koushika et al., 2001), and CLA-1S, a newly identified active zone protein homologous to vertebrate Piccolo and RIM (Xuan et al., 2017) in DA9. We found that UNC-10 and CLA-1S puncta regenerated only partially 48hr after axotomy. These new data are in contrast to our previous finding that regenerated SV clusters are restored to the pre-injury level (Figure 2E). Together, these data suggest that at least some of the new SV clusters are not fully restored synapses. This conclusion is consistent with our conclusion that the regenerated synapses are functionally weaker than those in unoperated controls. These new results are described in subsection “Formation of normal and dendritic synapses after regeneration of a single neuron” and in Figure 2—figure supplement 2. The new section reads:

“Normally, multiple protein components localize to active zones at the presynaptic terminal, where SV clusters are found. […] These data suggest that some of the new SV clusters in the regenerated axons represent defective synapses, and indicate that mechanisms for regenerating SV clusters and active zones are at least partially distinct.”

B) How are such SV clusters positioned relative to postsynaptic muscle receptor clusters? Do the muscle receptors stay in the same position during the entire period of axon degeneration and regrowth? Could some of the partial or aberrant behavior recovery be due to mis-alignment of new pre- and post-synaptic sites?

For postsynaptic markers (this is related to Essential revisions point 1A as well), we checked ACR-16 in the body wall muscles and ACR-12 in the GABAergic motor neurons. ACR-16 is a nicotinic acetylcholine receptor (nAChR) subunit in the muscles (Francis et al., 2005), and ACR-12 is a nAChR subunit expressed in the major classes of motor neurons (Petrash et al., 2013). Both ACR-16 and ACR-12 are expected to be postsynaptic to DA9 pre-synapses according to the anatomy and previous studies (Hall and Russell, 1991; Klassen and Shen, 2007; White et al., 1986). We found that both ACR-16 and ACR-12 puncta maintained their number and localization 48hr after axotomy in the DA9 synaptic region, suggesting that removing DA9 axon does not affect postsynaptic receptors.

To visualize co-localization, we labeled SV clusters in DA9 with BFP (Chai et al., 2012), ACR-12 in GABA neurons with GFP (Petrash et al., 2013), and DA9 with mCherry. BFP and GFP signals were analyzed for alignment of pre- and postsynaptic sites and mCherry was used to guide axotomy. We chose ACR-12 in GABA neurons rather than ACR-16 in body wall muscles as the postsynaptic marker, because its co-labeling with BFP::RAB-3 was significantly better in uninjured animals. In addition, ACR-12 labeling is more punctate compared to ACR-16, and so is more suitable for quantification of alignment. In brief, we found that the alignment of pre- and postsynaptic sites is restored after regeneration to the normal levels.

These new results are described in subsection “Postsynaptic receptors maintain their localization and are aligned with regenerated DA9 SV clusters after axotomy” and in Figure 3A-G and Figure 3—figure supplement 1A-D. The new section reads:

“Postsynaptic neurotransmitter receptors are normally juxtaposed to SV clusters and active zones in the presynaptic neuron, facilitating neurotransmission. […] Therefore, the alignment of pre- and postsynaptic sites is restored after axotomy and regeneration to normal levels.”

We also rewrote the last paragraph in the Discussion section (subsection “New synapses in the regenerating axon are not as efficient as intact ones”): “In our single-neuron system, although the regenerated synapses in the axon are able to drive some behavioral recovery, the level of dorsal tail bending is significantly lower than in intact controls. […] Our study highlights the difficulty in restoring normal circuit function after nerve injury, and provides insight into specific cellular choke points that need to be resolved to achieve complete recovery.”

C) Likewise, are the 'dendritic SV clusters' inducing new postsynaptic receptors, or altering existing postsynaptic sites? They showed ACR-2::GFP cluster was not affected by injury to axons. How are the injury-induced dendritic RAB-3 clusters formed in relation to ACR-2::GFP?

In the ACR-2 experiment, ACR-2 was labeled only in DA9, not in postsynaptic targets of DA9. This experiment was designed to test whether DA9 dendritic identity was maintained after axotomy.

To determine whether the dendritic SV clusters affect postsynaptic sites in the ventral body wall muscles, we labeled ACR-16 in the body wall muscles. We found that axotomy and regeneration of the axon do not induce significant changes in ACR-16 puncta. Unfortunately, co-labeling ACR-16 with GFP in the body wall muscles, RAB-3 with BFP in DA9 and DA9 axon with mCherry did not work (we need the third color to visualize the DA9 axon so we can cut it). So we could not analyze the colocalization between dendritic RAB-3 clusters and ACR-16::GFP in the ventral body wall muscles.

The new data are described in subsection “Postsynaptic receptors maintain their localization and are aligned with regenerated DA9 SV clusters after axotomy” and in Figure 3—figure supplement 1. The new section reads:

“To determine whether the new, ectopic dendritic SV clusters formed after axotomy affect postsynaptic receptors in the ventral body wall muscles, we analyzed the colocalization of ACR-16::GFP puncta in ventral muscle with the DA9 dendrite in the ventral nerve cord. […] Thus, the dendritic SV clusters that form in DA9 after axotomy do not induce significant numbers of new postsynaptic receptors in the ventral BWMs.”

2) Whether the tail-bending is mediated by chemical or electrical synapses:Is behavioral recovery and DA9 evoked muscle depolarization mediated by the regenerated synaptic puncta? To support this conclusion, the authors should determine if DA9 mediated behavior and muscle activation (GCAMP) following axotomy are blocked by mutations that block synaptic transmission (e.g. unc-13, unc-10, and unc-2).

We tested both behavioral responses and calcium signals in the *unc-13(e51)* background which blocks synaptic vesicle release (Richmond et al., 1999). We detected no behavioral responses or calcium signals in either intact or axotomized animals This suggests that the tail-bending behavior is mediated by chemical synapses in both intact and regenerated animals. The new data on behavior are described in subsection “Aberrant functional recovery of a single neuron-driven behaviour” and in Figure 4—figure supplement 2. This new section reads:

“To determine if the behavioral response is mediated by chemical or electrical synapses, we analyzed the behavior in *unc-13(e51)* animals in which synaptic transmission is blocked (Richmond et al., 1999). We detected no behavioral responses in either intact or axotomized animals 48hr after axotomy (Figure 4—figure supplement 2). This suggests that the tail-bending behavior is mediated by chemical synapses in both intact and regenerated animals.”

The new data on calcium imaging are described in subsection “Rerouted information transfer in a regenerated circuit” and in Figure 5—figure supplement 1. This new section reads:

“To determine whether the Ca^2+^ responses are mediated by chemical or electrical synapses, we examined Ca^2+^ signals in the *unc-13(e51)* background. No Ca^2+^ signals were detected in either intact or axotomized animals 48hr after axotomy (Figure 5—figure supplement 1CD). These data indicate that postsynaptic calcium responses due to DA9 stimulation are mediated by chemical synapses in both intact and regenerated axons, consistent with the results from behavioral analysis (Figure 4—figure supplement 2).”